# Effect of Different Configurations on Bubble Cutting and Process Intensification in a Micro-Structured Jet Bubble Column Using Digital Image Analysis

**Guanghui Chen \*, Zhongcheng Zhang, Fei Gao, Jianlong Li and Jipeng Dong \***

College of Chemical Engineering, Qingdao University of Science and Technology, Qingdao 266043, China; zzc19980124@163.com (Z.Z.); feigao@qust.edu.cn (F.G.); ljlong@qust.edu.cn (J.L.)
\* Correspondence: guanghui@qust.edu.cn (G.C.); dongjipeng0302@163.com (J.D.)

**Abstract:** An experimental study was conducted in this work to investigate the effect of different configurations on bubble cutting and process intensification in a micro-structured jet bubble column (MSJBC). Hydrodynamic parameters, including bubble size, flow field, liquid velocity, gas holdup as well as the interfacial area, were compared and researched for a MSJBC with and without mesh. The bubble dynamics and cutting images were recorded by a non-invasive optical measurement. An advanced particle image velocimetry technique (digital image analysis) was used to investigate the influence of different configurations on the surrounding flow field and liquid velocity. When there was a single mesh and two stages of mesh compared with no mesh, the experimental results showed that the bubble size decreased by 22.7% and 29.7%, the gas holdup increased by 5.7% and 9.7%, and the interfacial area increased by more than 34.8% and 43.5%, respectively. Significant changes in the flow field distribution caused by the intrusive effect of the mesh were observed, resulting in separate liquid circulation patterns near the wire mesh, which could alleviate the liquid back-mixing. The mass transfer experiment results on the chemical absorption of $CO_2$ into NaOH enhanced by a mass transfer process show that the reaction time to equilibrium is greatly reduced in the presence of the mesh in the column.

**Keywords:** bubble cutting; process intensification; mass transfer; micro-structured jet bubble column; bubble size distribution; chemical reaction engineering

## 1. Introduction

Bubble columns (BCs), as efficient multiphase contactors and bioreactors [1,2], are usually used in many industrial applications for gas–liquid contacting processes, especially in the case of a high interfacial area *a* and intense gas–liquid mixing [3,4]; however, further improvement of its structure and process optimization still faces great difficulties. The main limiting step for an efficient reactor operation in chemisorption processes such as in tail gas treatment and wastewater treatment is the mass transfer efficiency (e.g., the overall gas holdup $\varepsilon_g$ and mixing time) of gas–liquid phases [5], namely, the reactor performance.

In large-scale industrial production, the inlet gas velocity is often very fast, which manifests as a jet, forming a group of adhesive bubbles in the column (jet flow). Jet injection motivates complex and unsteady flow regime conditions. Understanding the flow and mixing characteristics of impinging jets is of great significance for engineering applications. The impingement of the jet flow leads to the turbulence of the fluid greatly increasing, and the diffusion path is reduced, thereby achieving a rapid and effective mixing [6]. The jet also brings a huge time-averaged gas intake, and frequent bubble collision is the main factor causing coalescence; however, in these processes, the bubble coalescence causes that the interfacial area and the contact area between the gas and liquid phases to be greatly restricted.

To overcome these drawbacks and limitations caused by the bubble coalescence, numerous methods for lessening bubble size have been developed to increase the gas–liquid interfacial area and mass transfer coefficient. The mass transfer rate can be increased by increasing the interfacial area $a$ and the volumetric mass transfer coefficient kl$\alpha$ [7]. The reduction in the mean bubble size can be roughly divided into two scenarios. One is that the bubble was already a relatively stable, small bubble at the beginning of its formation [8], and these bubbles seldom burst while usually interacting and coalescing with the surrounding bubbles. The other scenario is where large bubbles break into daughter bubbles due to their own or environmental restrictions (e.g., gas and liquid properties, environmental temperature, operating pressure, particle concentration, liquid level height, etc.) or external forces (e.g., bubble collision, liquid turbulence, local eddy, etc.) during the rising process. Turbulence, including bubble-induced turbulence [9–11], and a local flow eddy [12] could be regarded as an important factor in improving the mass transfer and bubble breakup rate in the BCs, since strong turbulence characteristics [13] and eddy can improve the poor mixing capacity in a gas–liquid flow.

The internals, as an intrusive device, equipped in BCs have been perceived as an effective method for achieving mass transfer intensification [14–16]. Researchers have found that enhancing bubble coalescence and breakup processes in the liquid phase can promote mass transfer [17], and the results show that the significant crests in the mass transfer coefficient coincide with significant crests in the bubble breakup and coalescence rates. Bubble cutting is broadly considered as a constructive method to achieve the purpose of process intensification by imposing external forces to cut large bubbles into sub-bubbles, limiting the growth of bubble size and aggrandizing the breakup rate of the bubbles. In our previous work [18,19], a better mass transfer efficiency was achieved by installing wire mesh inside the column to break large bubbles into small ones, and it was found that the smaller bubble size generated by the cutting could lead to a lower rising velocity, larger gas holdup, larger gas–liquid interfacial area, a more efficient gas diffusion rate and a larger phase interface renewal rate. In another work of ours [20], the mass transfer conditions in a $CO_2/N_2$-water system were taken into consideration and the enhancement effect of cutting behavior on the mass transfer process was verified systematically based on the combination of numerical simulation and a custom user-defined function (UDF) instruction. The interaction between the rising bubble and the mesh enhances the bubble dynamics, improves the poor performance of the gas–liquid two phases and increases the local mass transfer coefficient [21,22]. The introduction of the wire mesh causes a significant intrusive effect in the column such as bubble deformation, breakup, deceleration, turbulence, local vortex and liquid circulation. Sujatha et al. [23] found that the inserted wire mesh changed the trajectory of the bubble, aggravated the local fluid turbulence, and disrupted the normal development of the bubble plume. The catalyst could also be loaded on the wire mesh [24–26], reducing the cost of filtering and recovering the catalyst particles.

In the present work, the main aim was to extend the work on the effect of different mesh configurations on bubble cutting and flow characteristics in a novel micro-structured jet bubble column (MSJBC). The gas–liquid flow characteristics in the wide range of high inlet velocity (140~570 cm/s) were studied using digital image analysis. To investigate the effect of the internal components on the bubble size, nitrogen was used as the gas phase in the experiment to avoid measurement deviation of the overall bubble size caused by the dissolution of the gas into the liquid phase. The intensification of the mass transfer process by a varying number of internal stages (no wire mesh, single stage mesh, two stages mesh) in the MSJBC was verified by chemical adsorption experiments of $CO_2$ into NaOH. Through detailed investigation and comparison of the bubble dynamics parameters including bubble size number density function (NDF), Sauter mean bubble diameter $d_{32}$ and gas–liquid interfacial area $a$; and fluid parameters including pH, gas holdup $\varepsilon_g$, flow field and liquid velocity, the influence of internals and gas velocity changes were analyzed. A non-intrusive high-speed camera method was adopted for the characterization of the

bubble morphology, bubble size and size distribution, and particle image velocimetry (PIV) was adopted for the flow field and liquid velocity vector.

## 2. Experimental Setup

### 2.1. Experimental Devices and Flow Conditions

The experimental flow-process diagram is illustrated in Figure 1. The experimental devices consisted of a rectangular MSJBC (with dimensions of 100 mm width, 25 mm depth and 700 mm height) and one or two wire meshes (with 5.5 mm mesh opening and 0.7 mm wire diameter). In order to better study the bubble morphology and size distribution after bubble cutting, a rectangular MSJBC was made of transparent acrylic material. Rectangular structures were suitable for the lab-scale research and have been used in many researches [27–30]. When capturing the bubble images with the high-speed camera, rectangular BCs are easier to visually observe and optically measure than cylindrical structures [31] because the curved side walls of the cylindrical structures have a strong divergence effect on laser light. The gas was fed into the column through a gas nozzle arranged on the bottom plate in the center. Needle length = 50 mm, inner diameter = 1 mm and outer diameter = 1.5 mm. The needle extended 20 mm above the base plate. The material of the wire mesh was 304 stainless steel and the installation position of the wire mesh was fixed 175 mm (lower wire mesh) and 225 mm (upper wire mesh) away from the bottom single gas nozzle, respectively. The use of a flow meter to control the gas flow rate, allowed the flow rate to vary from 36 to 144 l/h. The corresponding superficial gas velocities $U_g$ and inlet velocity are shown in Table 1.

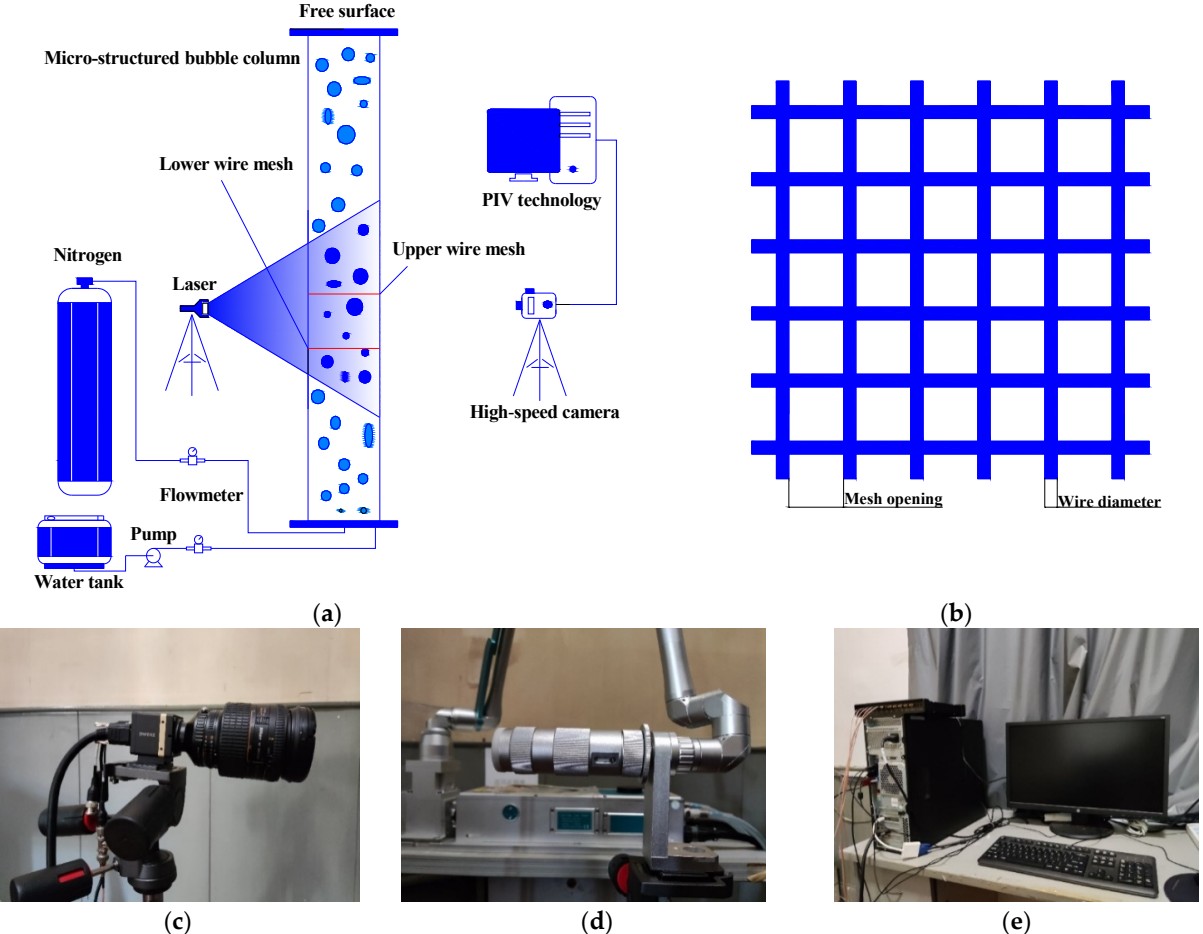

**Figure 1.** Schematic of the experimental devices: (**a**) experimental system, (**b**) wire mesh, (**c**) high-speed camera, (**d**) laser and (**e**) PIV.

Table 1. Flow conditions in this work.

| Cases | 1 | 2 | 3 | 4 |
|---|---|---|---|---|
| Gas flow rate (l/h) | 36 | 72 | 108 | 144 |
| $U_g$ (mm/s) | 4 | 8 | 12 | 16 |
| Inlet gas velocity (cm/s) | 141.5 | 283 | 424.5 | 566 |

### 2.2. Materials

Nitrogen (99.999%) and deionized water were selected as the gas and liquid phases, and can be considered an incompressible fluid, respectively. Nitrogen is insoluble in water and the bubble shrinkage caused by the dissolution of nitrogen in the liquid phase can be ignored. The chemical absorption and mass transfer experiments in the current research work were set as a $CO_2$-NaOH aqueous solution system. The consecutive reversible reaction equation is as follows:

$$CO_2(g) \rightarrow CO_2(aq) \tag{1}$$

$$OH^-(aq) + CO_2(aq) \rightleftharpoons HCO_3^-(aq) \tag{2}$$

$$HCO_3^-(aq) + OH^-(aq) \rightleftharpoons H_2O(l) + CO_3^{2-}(aq) \tag{3}$$

where Equation (1) is the physical absorption process of $CO_2$ into the liquid phase, Equations (2) and (3) indicate that dissolved $CO_2$ combines with hydroxide ion ($OH^-$) to form bicarbonate ion ($HCO_3^-$) and carbonate ion ($CO_3^{2-}$).

In the early stages of a continuous reaction system, Equation (3) dominates the reaction. The pH value of the solution rapidly descends with the consumption of $OH^-$ and the accumulation of $CO_3^{2-}$ as the reaction progresses. The total absorption of the $CO_2$ at the initial stage of the reaction can be expressed as:

$$2NaOH(aq) + CO_2(aq) \rightleftharpoons Na_2CO_3(aq) + H_2O(l)$$

It can be seen from the chemical equilibrium that the increase in carbonate concentration promotes the positive reaction of Equation (2) and the backward reaction of Equation (3), both of which promote the formation of $HCO_3^-$. Thus, the pH value of the solution decreased slowly in the middle and late stage of the reaction. The total absorption reaction of the second stage can be expressed as:

$$Na_2CO_3(aq) + CO_2(aq) + H_2O(l) \rightleftharpoons 2NaHCO_3(aq)$$

In all cases, the initial solution pH was set to 12.58 and the liquid layer height was 600 mm. Phenolphthalein was dropped into the sodium hydroxide solution to facilitate visual observation of the chemical reaction process before the $CO_2$ was introduced.

## 3. Measurement Methods

### 3.1. Capture Bubbles

A non-intrusive optical method, namely a high-speed camera method, was adopted for the experiment. Its characteristics such as simplicity and physical intuition are attractive in laboratory-scale research. Non-intrusive methods, such as the optical method and acoustic bubble detection method [32], are superior to intrusive ones (e.g., optical fiber probe, conductivity probe and wire mesh sensor), due to (1) the former realizing the combination of PIV technology and visualization observation; (2) bubbles are not idealized rigid spheres, and intrusive measurements are subject to measurement biases [27]; and (3) the latter can disturb the fluid flow when inserted into the column [33]. The bubble information provided by the high-speed camera can be obtained by physical visual observation and post processing technology (PIV). The high-speed camera method could provide a clear gas–liquid interface, but this method is only suitable for low gas fraction conditions, and a

sufficient optical accessibility is essential. The MSJBC was divided into four measurement zones, as shown in Figure 2. The measurement zones were maintained for the same initial liquid heights ($H_0$ = 600 mm). Images of the bubbles were captured at a rate of 50 Hz by a high-speed camera. The bubbles were monitored and captured by a high-speed camera and laser light moving to the corresponding column height. The camera position was set perpendicular to the wire mesh. To obtain as much bubble size as possible and to ensure enough bubble images were captured entering and leaving the column without being counted repeatedly, 20 images of recognized bubbles were taken for each measurement zone according to the recorded time series.

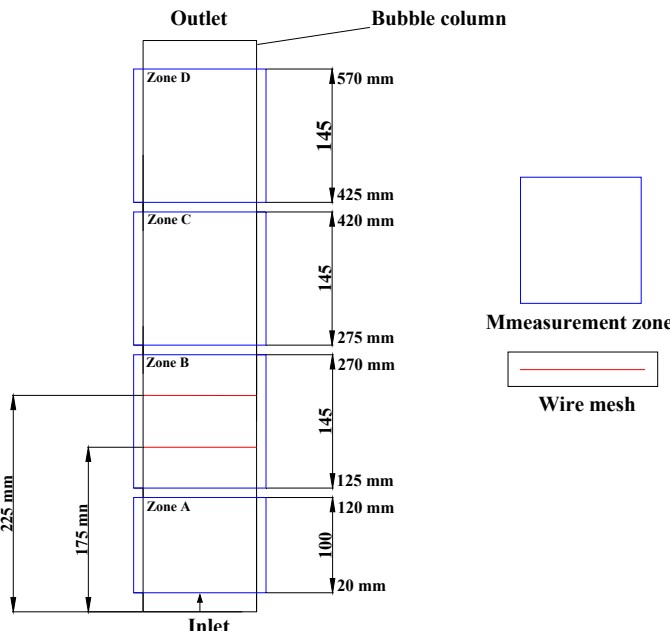

**Figure 2.** Division of different measurement zones (Zone A, Zone B, Zone C and Zone D) in a MSJBC.

### 3.2. Bubble Size Measurement

When the bubble size is large, the bubble is often not round but deformed. A simple and practicable method, using an equivalent bubble diameter $d_{eq}$ to replace the diameter of each deformed bubble, has been widely used in chemical engineering and related fields. The calculation method is as follows [34]:

$$d_{eq} = \sqrt[3]{6V_b/\pi} \tag{4}$$

$$V_b = \left(\pi C_1^2 \cdot C_0\right)/6 \tag{5}$$

where $C_1$ is the major axis (chord length) of the ellipse bubble and $C_0$ is the minor axis (chord length) of the bubble. A transparent scale is needed to calibrate the camera's focus location and determine the length of the unit pixel (see Figure 3).

The bubble Sauter mean diameter $d_{32}$, as one of the key parameters of bubble dynamics, determines the mass transfer process performance of the gas–liquid contacting equipment since the interfacial area depends largely on $d_{32}$. The experimental data of $d_{32}$ can be calculated through the following formula in this work:

$$d_{32} = \sum_{i=1}^{N} d_{eq,i}^3 / \sum_{i=1}^{N} d_{eq,i}^2 \tag{6}$$

The gas–liquid interfacial area $a$ is a function of the gas holdup and $d_{32}$, which can be calculated by the following formula:

$$a = 6\varepsilon_g / d_{32} \tag{7}$$

The gas holdup is determined for the gas–liquid system by the expansion of the liquid height and calculated by the following formula:

$$\varepsilon_g = \left(h_f - h_i\right)/h_i \tag{8}$$

The number density function (*NDF*) for each bubble diameter $d_{eq,i}$ is calculated from the number of bubbles and $d_{eq}$:

$$NDF_{d_{eq,i}} = N_{d_{eq}} / \sum\nolimits_{d_{eq,min}}^{d_{eq,max}} N_{d_{eq,i}} \tag{9}$$

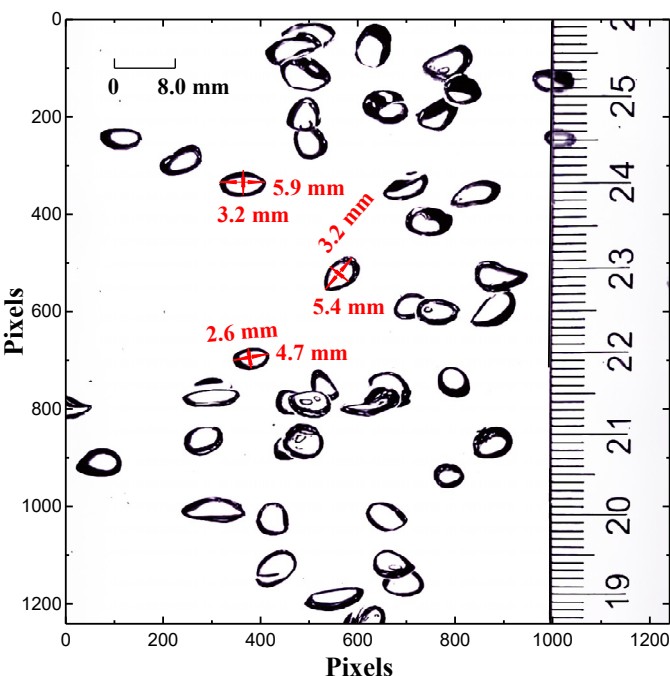

**Figure 3.** Calibrating the focus position and measuring the length of the unit pixel (0.064 mm/pixel).

### 3.3. Liquid Velocity and Flow Field

Continuous bubble images were recorded by a high-speed camera and processed by PIV technology using digital image analysis to obtain the liquid velocity $\bar{v}$ and flow field. Laser-induced fluorescence technology (LIF) was used to process the flow field with tracer particles irradiated by laser. The tracer particles with good following and fluidity were added into the liquid phase. When the laser generator emits a slice light source with an appropriate frequency to illuminate the tracer particles, the tracer particles are illuminated and the energy-absorbing tracer particles will reflect a longer fluorescence wavelength than the laser. The brightness of the tracer particles is in sharp contrast with the surrounding flow field to obtain the flow field information through computer processing. The instantaneous flow field was photographed by the image capture equipment (high-speed camera) at the same frequency as the laser to obtain the flow field images at different times. The particle displacement $\Delta \overline{L}$ was calculated according to the correlation detection of particles in each infinitesimal in the image at different times. Combined with the recording time interval $\Delta t$, the movement velocity of tracer particles, namely the liquid flow velocity of fluid, was calculated (see Equation (8)). In this work, hollow glass spheres (density 1055 kg/m$^3$ and diameter 10 μm) were selected as the tracer particles required by the experiment after comprehensive consideration of the tracer's following, fluidity and agglomeration phenomenon in the liquid. The density of hollow glass spheres is similar to that of the liquid phase, and they are easier to suspend in the liquid phase, which can give full play to

the tracer effect. The particle velocity and trajectory were most consistent with the actual flow field. The processing process of the liquid velocity and flow field is shown in Figure 4.

$$\overline{v} = \Delta\overline{L}/\Delta t \tag{10}$$

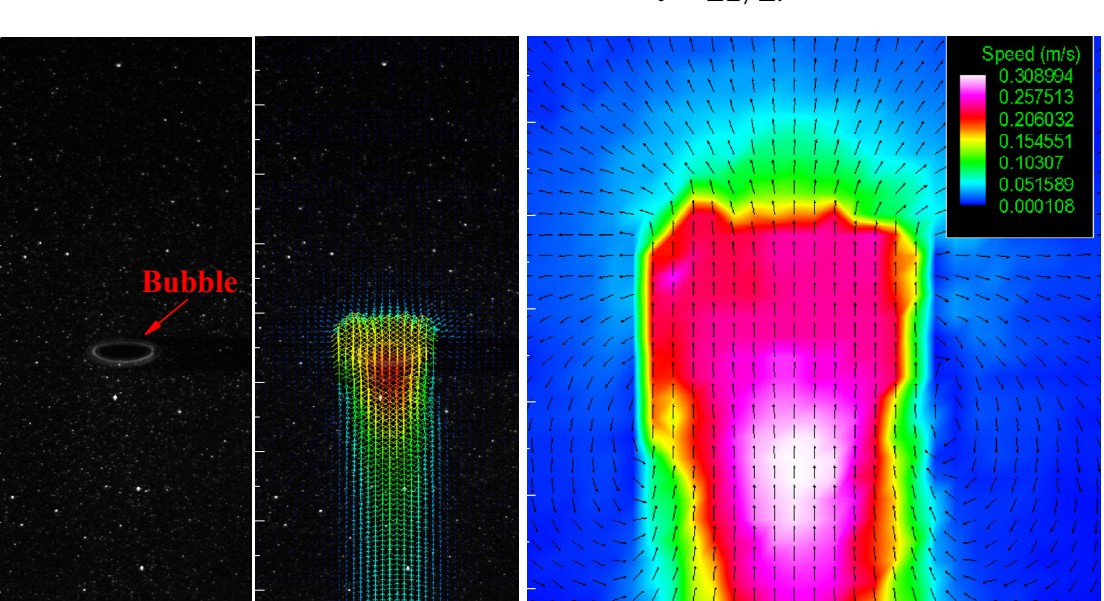

(**a**)　　　　　　　　(**b**)　　　　　　　　(**c**)

**Figure 4.** Liquid velocity measurement: (**a**) the original recorded image; (**b**) velocity image pre-processing; (**c**) flow field and velocity vector of bubbles post-processing.

## 4. Results and Discussion

### 4.1. Effect of Different Configurations on Bubble Morphology

Bubble images were recorded using the high-speed camera operated at 50 Hz with superficial gas velocities ranging from 4 to 16 mm/s. It should be noted that these snapshots of the recorded bubble sequences are for visual analysis and do not adequately and intuitively illustrate the bubble dynamics. The preliminary visual observation of the effect of internals (wire mesh) on bubble size and morphology is shown in Figure 5 and the black horizontal shadow is the projection of the front view of the wire mesh, with the following results being observed. The gas was injected in the form of a jet, and the gas velocity at the entrance was too fast to form a gas bundle. As the gas bundle developed, it gradually disintegrated into a string of upward moving bubbles. In the process of transforming from a gas bundle to bubble cluster, bubbles were broken due to the impact of a gas bundle fracture and turbulence, thus forming an independent bubble cluster. Such a small bubble cluster usually had 2~4 leading bubbles in the front, and the smaller bubbles formed by bursting under the wake capture effect of the leading bubble, as shown in Figure 5a. As shown in Figure 5a–c, the bubble cutting occurred in the presence of the mesh. As shown in Figure 5d,e, the leading bubbles were effectively cut into numerous daughter bubbles, and the daughter bubbles were dispersed due to the invasion effect of the wire mesh, avoiding the recoalescence during the ascent process. The equivalent size of the leading bubble ranged from 15 to 20 mm, but the bubble size decreased sharply after cutting, and most of the bubbles' size was between 2 and 8 mm.

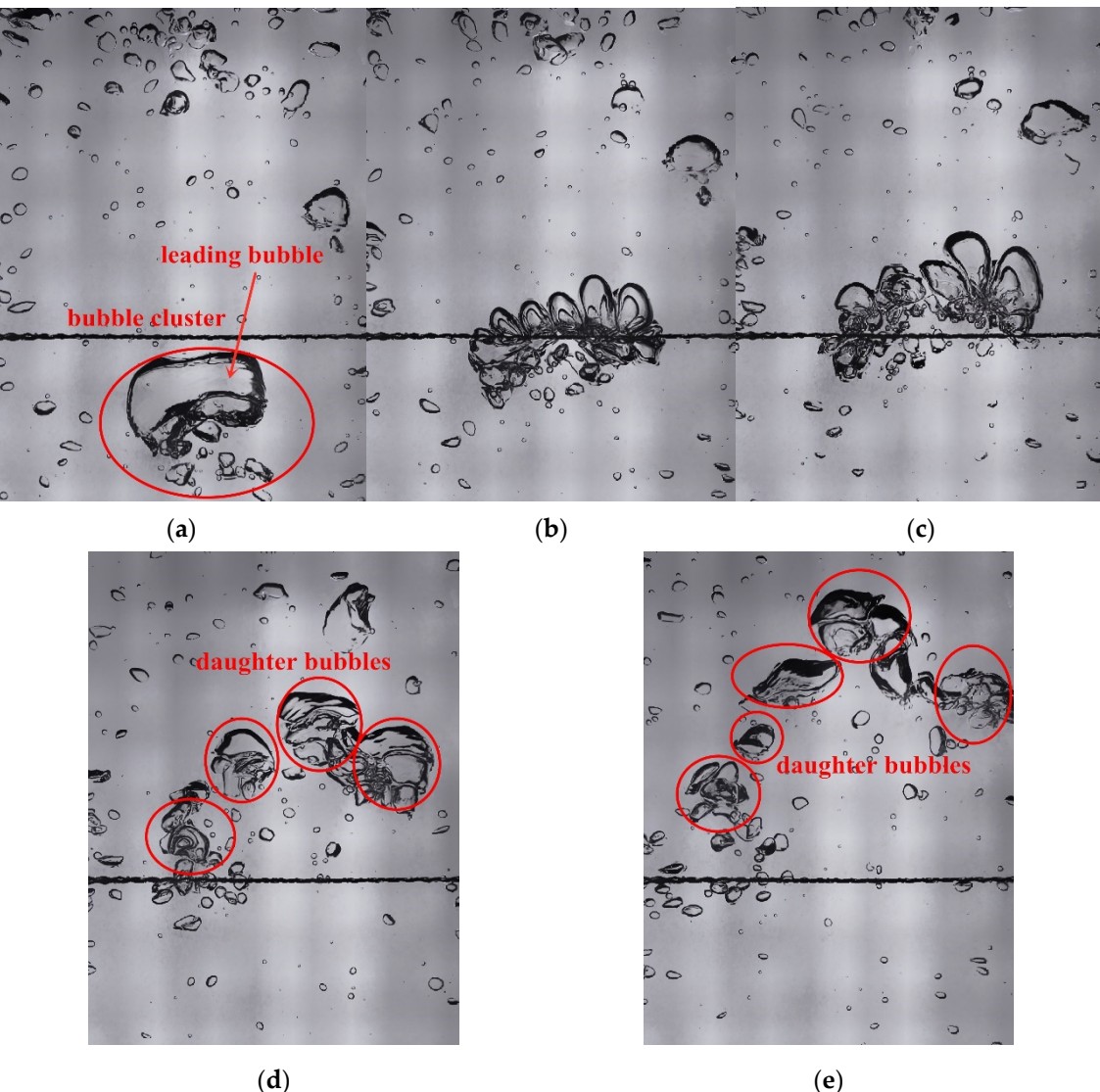

**Figure 5.** Bubble cutting behavior with single wire mesh: daughter bubbles' formation occurs after cutting at a superficial of 4 mm/s. (**a**) represents the uncut leading bubble, (**b**) represents the leading bubble is cut, (**c**) represents the leading bubble has been successfully cut, (**d**) represents the leading bubble has been cut into daughter bubbles and (**e**) represents the dispersed daughter bubbles.

Figure 6 denotes the bubble cutting–recoalescence phenomenon during the cutting progress. This phenomenon often occurs in the case of a bubble cluster with multiple leading bubbles. The leading bubbles stacked on top of each other under the influence of the wake effect and different rising velocity (see Figure 6a). In view of the mesh intrusive effect, the first bubble gradually approached the mesh subject due to the deceleration effect, and the ascent speed decreased immediately and rapidly. Due to the difference in rising velocity, the lower bubbles collided with the decelerated bubbles and coalesced to form larger aggregation bubbles (see Figure 6b). Large aggregation bubbles in the cutting process were not completely cut, and part of the bubbles passing through the wire mesh did recoalescence (see Figure 6c–e). In the end, all the bubbles were detached from the mesh, but the bubble size was almost the same. In some cases, this cutting phenomenon resulted in the rupture of the tail of the large bubble, resulting in the formation of several sub-bubbles.

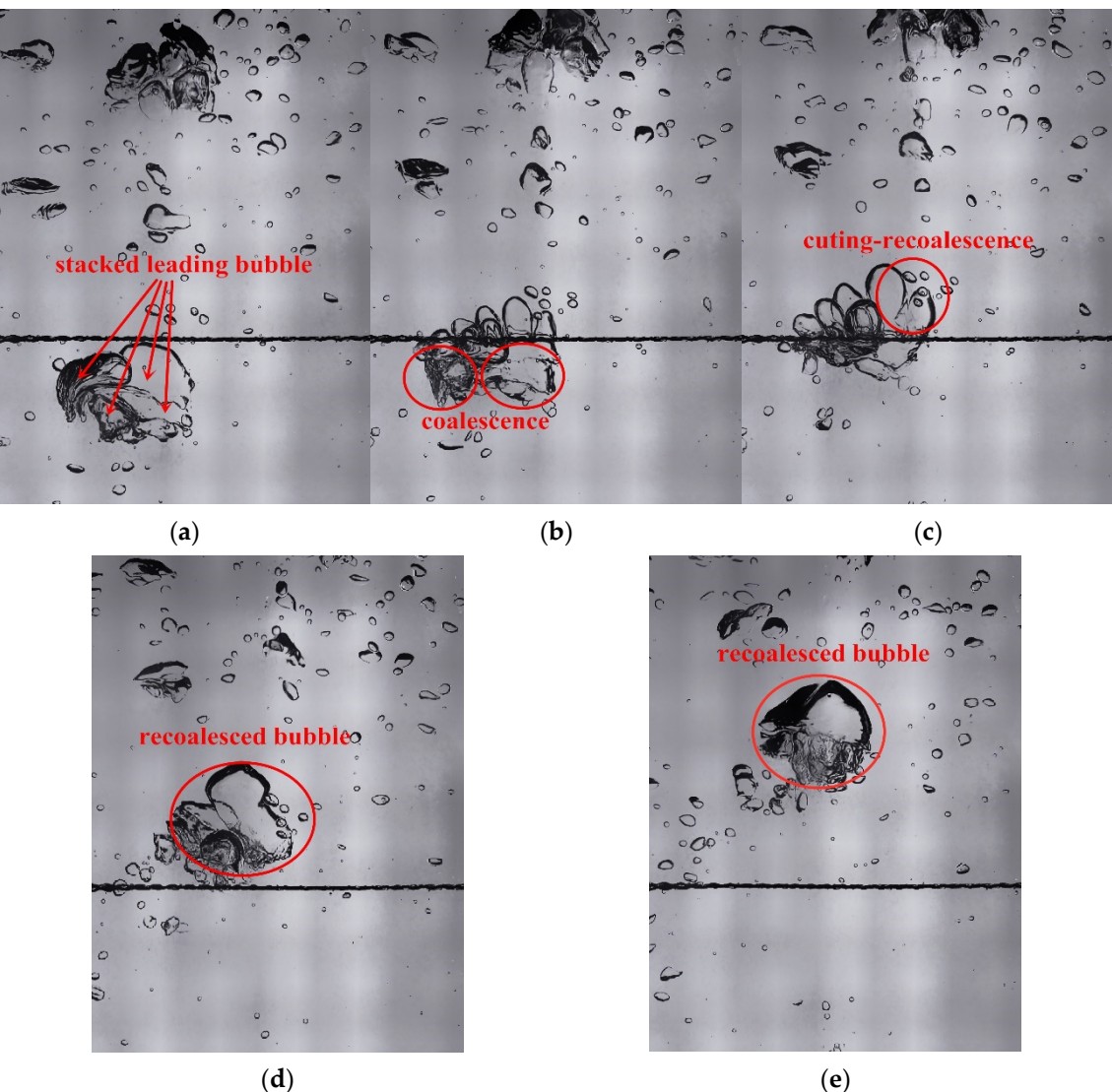

**Figure 6.** Bubble cutting behavior with single wire mesh: bubble cut–recoalesce occurs during the cutting process at a superficial of 4 mm/s. (**a**) represents the stacked leading bubbles, (**b**) represents the coalescence of the leading bubbles, (**c**) represents the cut-recoalescence phenomenon of the leading bubbles, (**d**) represents the recoalescence of the leading bubbles and (**e**) represents the recoalescence of the leading bubbles.

Bubble re-cutting behavior occurred in the presence of the two stages mesh (see Figure 7). Bubbles that passed through the lower wire mesh were not cut or were not effectively cut (see Figure 7c) and could be cut again when passing through the upper wire mesh (see Figure 7e). The two-mesh structure increased the possibility of bubbles bursting. In addition, the two stages wire mesh inserted into the column had more intrusive effects on the flow field than the single-stage. A unique flow field formed between the two wire meshes (we will discuss this in Figure 9), and small bubbles were captured by small vortices and it was difficult to escape in these flow field conditions. The change in the flow field before and after cutting will be discussed in Section 4.2. For the discrete bubbles successfully cut by the lower mesh, the upper mesh could also assume the function of a gas distributor, which increased the spacing of bubbles, and reduced the bubble coalesce caused by the frequent collisions of bubbles.

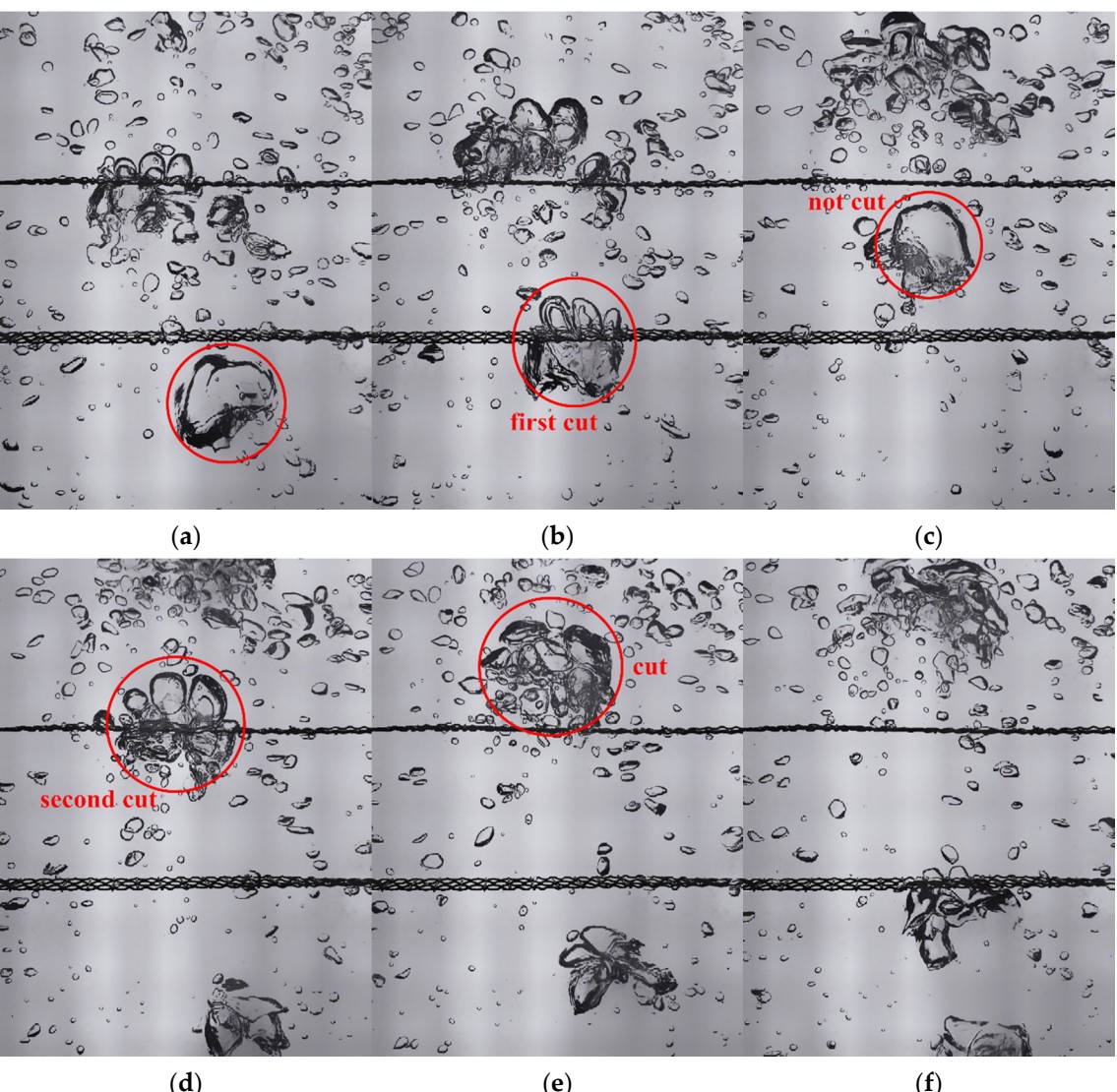

**Figure 7.** Bubble re-cutting behavior with the two stages wire mesh: bubbles that are not cut by the lower mesh but are cut by the upper mesh at a superficial of 4 mm/s. (**a**) represents the large bubble that has not been cut, (**b**) represents the bubble that has been cut for the first time, (**c**) represents the bubble that has not been cut by the lower mesh, (**d**) represents the second cut of the bubble, (**e**) represents the bubble that has been successfully cut by the upper mesh, and (**f**) represents the daughter bubble that has been cut.

### 4.2. Effect of Different Mesh Configurations on the Flow Field

In this section, the intrusive effect of the inserted wire mesh on the surrounding flow field was examined. In Figure 8, the red horizontal dashed line indicates the location of the wire mesh. The instantaneous flow field distribution obtained after post-processing is shown in Figure 8. Figure 8a,c are the original images, while Figure 8b,d are the flow field information images processed in Figure 8a,c respectively. From Figure 8d,e, the distribution of the flow field was scattered and chaotic, indicating that the gas at a low superficial gas velocity did not play a role in regulating the flow field. The flow field distribution around the cut bubbles was divergent, which could be devoted to dispersing bubbles, increasing the distance between bubbles and reducing the coalescence rate of the bubbles. There were small eddies resting in the path of the bubbles. The dissipation of a small vortex resulted in a disorderly change in the flow field. The flow field around the bubbles after cutting produced local vortices under the influence of discrete bubbles. In the presence of the two stages mesh, the intrusive effect of the wire mesh increased, the flow field suffered stronger

interference, and the chaotic degree of the flow field increased. Some small local vortices were formed between the mesh.

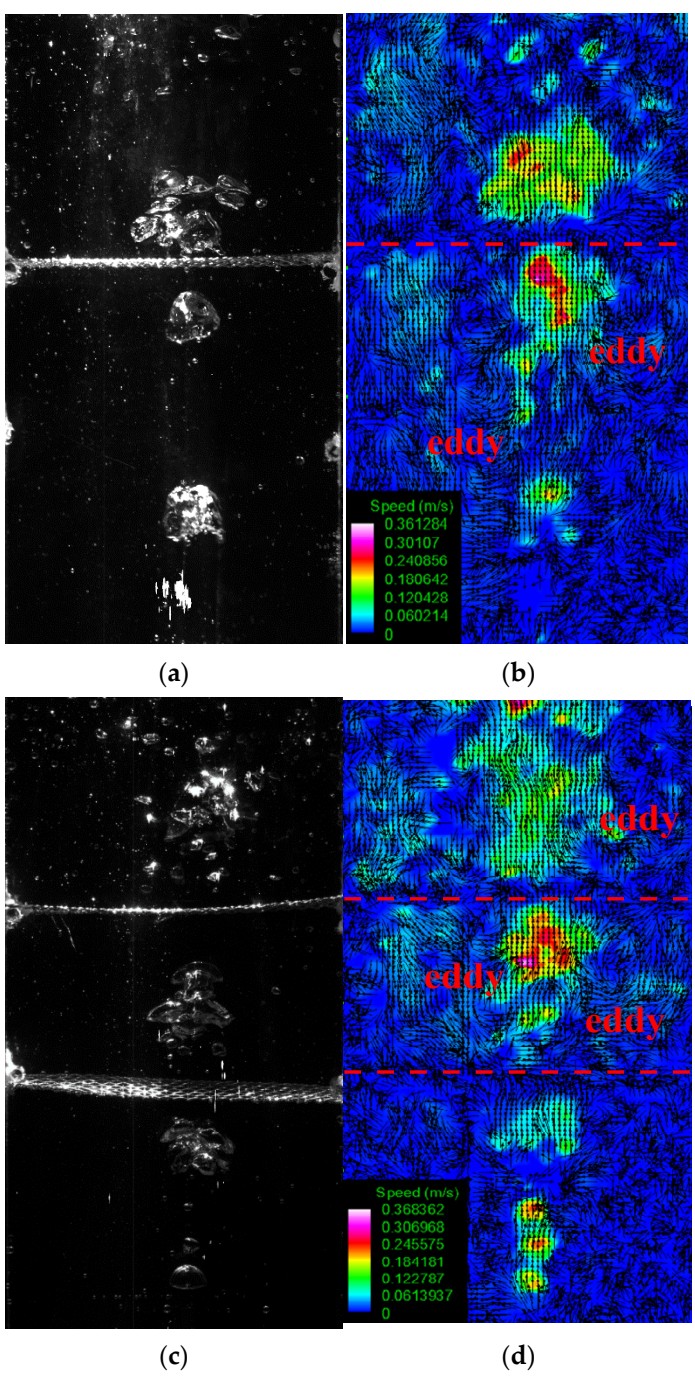

**Figure 8.** Surrounding flow field distribution for bubble cutting at a superficial of 4 mm/s. Upper row: single wire mesh. Lower row: two stages mesh. Original images: (**a**,**c**); Processed images: (**b**,**d**).

Figure 9 shows the effect of the mesh structure inserted into the column on the global flow field distribution, and the superficial gas velocity selected as 12 mm/s. From Figure 9, according to the images of the flow field in a MSJBC for no stage mesh compared with different configurations, a significant difference in the formation of an independent liquid circulation pattern was observed. Figure 9a shows the instantaneous flow field recorded without the wire mesh insertion. The gas flow trajectory was the classical "S" shape, and two large-scale vortices and liquid backflow were formed on the concave surface of the S-shape trajectory. Figure 9b shows the distribution of the instantaneous flow field with

the single structure wire mesh. Compared with no wire mesh, wire mesh with a single structure in the column mainly affected the flow field distribution below. The flow field distribution underneath the mesh with backflow pattern was destroyed, but there was still a large-scale vortex above the mesh.

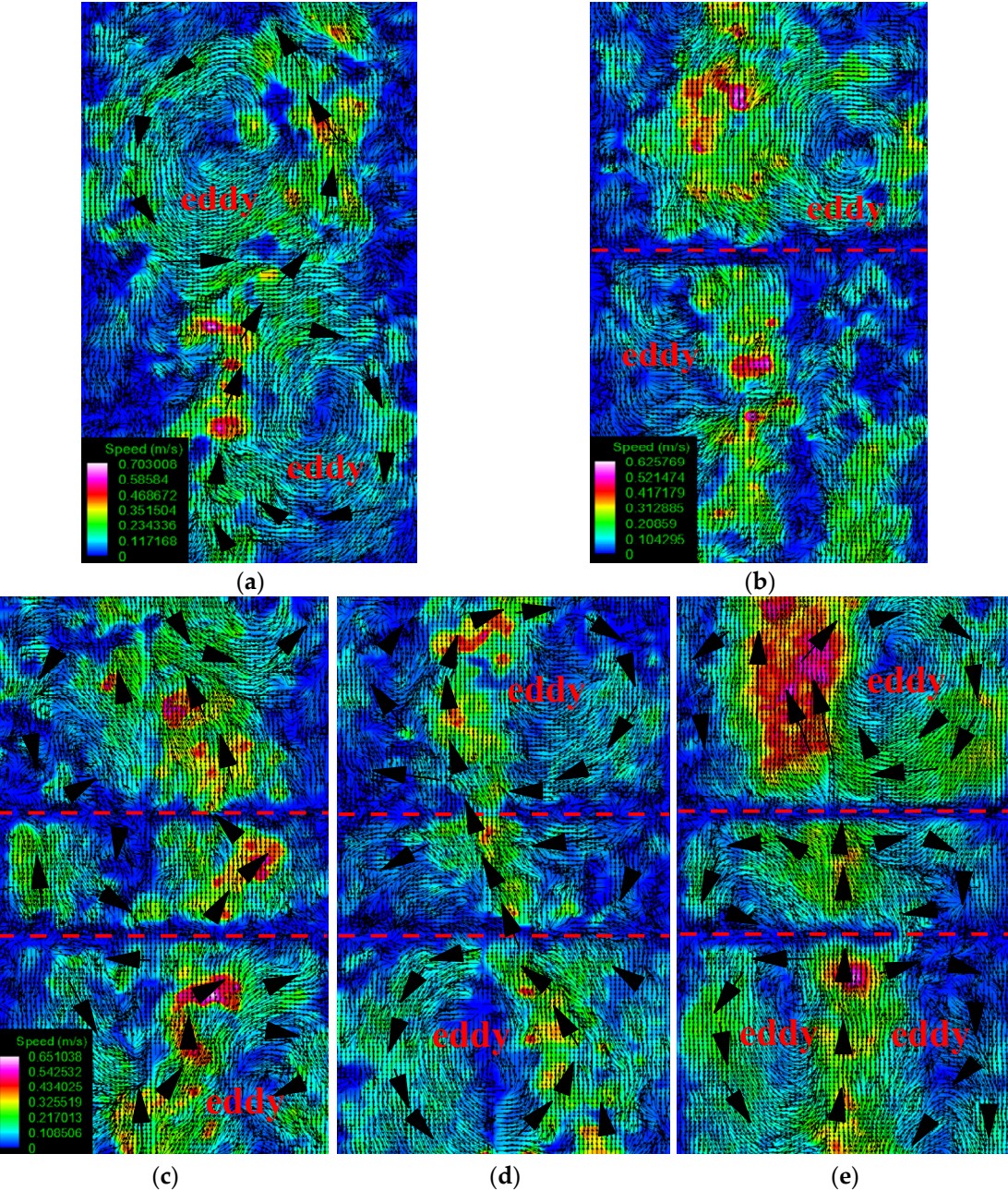

**Figure 9.** Instantaneous flow field and corresponding liquid velocity distribution at a superficial velocity of 12 mm/s in the height range of 85 mm to 315 mm. Upper row (**a**): no wire mesh. Upper row (**b**): single wire mesh. Lower row (**c**–**e**): two stages mesh.

As shown in Figure 9c–e, three representative flow field distributions were obtained in the experiment when the superficial gas velocity was 12 mm/s. Figure 9c shows a haphazard flow field distribution, characterized by chaos and disorder, which only appeared when the degree of chaos in the fluid flow exceeded a certain limit. The jet caused the violent turbulence of the fluid, which in turn affected the stability of the jet and deepened the chaos of the flow field. In Figure 9d, a great influence of liquid reflux on

the flow field was observed. The backflow effect of the lower mesh impacted the rising path of the bubble. The curvature of the bubble trajectory affected the flow field through a feedback mechanism, causing changes in the flow field below, resulting in a closed circular distribution of the flow field and because of the influence of the curved bubble trajectory, the bubble was not perpendicular to the mesh but rather with a certain angle. The bubble cutting caused by this non-vertical form through the mesh increased the radial movement of the bubble, which was not conducive to maintaining the stability of the flow field. Above the upper mesh, the circular flow field appeared. Figure 9e represents the third flow field. A great characteristic of jet streams is that they tend to form a straight gas beam. High-speed gas beneath the upper mesh split the flow field in two. The high speed gas from the center of the bottom plate hit the wire mesh vertically, and the development of a liquid recirculation loop was inhibited due to the liquid being subject to a blocking action. The higher the flow rate, the lower the pressure. Part of the descending fluid was sucked into the main flow channel under the influence of the pressure difference. The flow field above the upper wire mesh was similar to the distribution demonstrated in Figure 9d. Since the strong liquid circulation existed in the MSJBC, the bubble flow was quite different from the fluid flow with no mesh.

The gas plume was a dynamic region within the column in which the bubbles rose at a higher velocity. This dynamic region had larger bubbles due to the merger of bubbles. Smaller bubbles gathered near the side walls and were dragged down by the flow of the liquid. In the flow field demonstrated in Figure 9, the intrusive effect of the wire mesh resulted in the formation of independent liquid circulation patterns for the sections above, middle and below the mesh (which interfered with plume development), thereby reducing the liquid back-mixing inside the column. The downward flow of small bubbles was clearly visible in visual observation. The residence times of the small bubbles were significantly prolonged due to the strong liquid circulation, which made it difficult for these bubbles to escape the column.

*4.3. Effect of Different Mesh Configurations on Bubble Size*

The bubble shrinkage effect caused by the dissolution of nitrogen in the liquid phase was extremely weak and negligible. In all investigated cases, the bubbles' natural breakup and cutting was the main factor leading to the dispersed bubble size distribution. Bubble growth was mainly induced by bubble coalescence. The NDF of each bubble size for the different superficial gas velocities is illustrated in Figure 10. In Figure 10, the histogram represents the bubble size distribution (NDF), and the curve represents the cumulative frequency of bubble size. The same colors indicate the same mesh configuration, respectively.

By observing the absence of the wire mesh in Figure 10, the superficial gas velocity (or inlet velocity) had a dual effect on bubble size distribution under jet flow conditions. An increment in gas velocity increased the percentage of large-size bubbles ($\geq$15 mm), but the impact of a single-hole jet also strengthened the fluid turbulence, thus increasing the bubble breakup frequency and driving the increase in the proportion of small bubbles. The turbulence caused by excessive gas velocity was more severe, which was not conducive to the stable existence of large bubbles. It can be seen from Figure 10 that the most significant change in the presence of the wire mesh was that to the peak of the bubble size distribution which shifted to a smaller size (left). The proportion of bubbles with a size less than 5 mm increased, while the proportion of bubbles with a size greater than 15 mm decreased or even disappeared due to the cutting capacity and intrusion effect of the wire mesh. By analyzing the NDF distribution of the single stage and two stages mesh, combined with the visual observation in Section 4.1, the two stages mesh had a higher cutting capacity and efficiency. Compared with the case of no mesh, the proportion of bubbles sized smaller than 5 mm in the presence of the two mesh configurations increased by 49%, and the proportion of bubbles with the single stage mesh increased by 22.5%. The mesh insertion with a composite structure improved the "cutting fault tolerance" of the single structure mesh. Another reason for the increased proportion of small bubbles can be attributed to

the independent circulation pattern of the flow field caused by the inserted wire mesh, which caused the small bubbles to remain in the MSJBC for a long time.

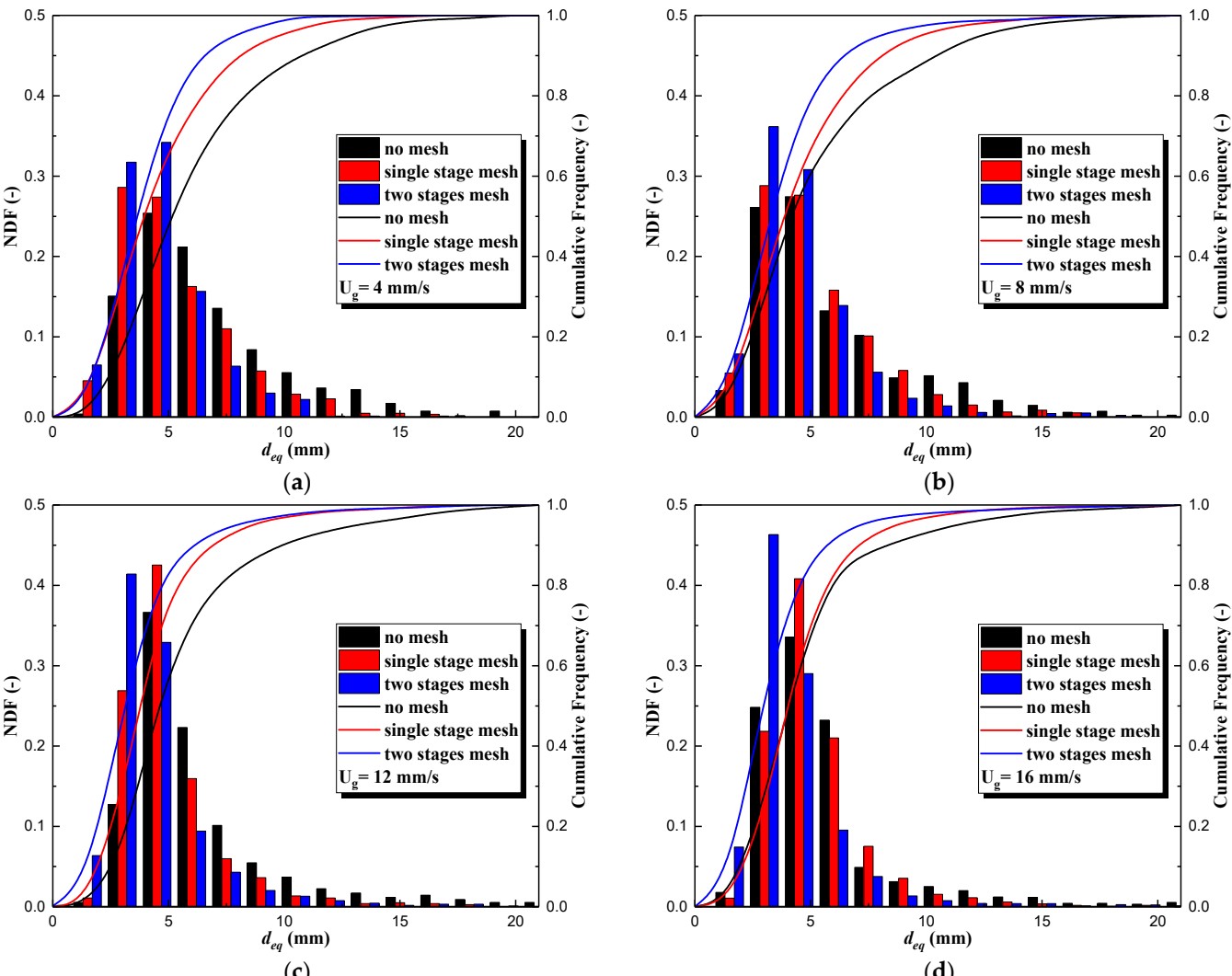

**Figure 10.** Evolution of bubble size distributions at different superficial velocity equipped with no, single stage and two stages mesh without chemisorption. (**a**–**d**) represent the bubble size number density distribution in MSJBC when the superficial gas velocity is 4, 8, 12 and 16 mm/s, respectively.

The excellent performance of the two stages mesh was also reflected in the cutting efficiency of the high superficial gas velocity. The cumulative frequency curve represents the cumulative amplitude of the bubble number density function, and the fastest growing bubble size proportion can be intuitively observed through the slope of the curve. From the cumulative curve in Figure 10, in the prophase of the cumulative curve, the steepest slope is for inserting the two stages mesh (blue), followed by the single stage mesh (red), and the case without mesh (black) shows the slowest growth. The red curve has a tendency to move closer to the black curve with the increasing gas velocity, indicating that the growth of gas velocity weakened the cutting capacity of the single wire mesh, and that the bubbles were easier to pass through the mesh without being cut (see Figure 6).

Figure 11 shows the change in the Sauter mean diameter as a function of column height. In Figure 11, the black square represents the size change without the mesh, the red triangle represents the size change only through the lower wire mesh (red vertical line), and the blue diamond represents the size change through the lower and upper wire mesh (black vertical line). From Figure 11, it can be seen that in the absence of a wire mesh

(black square) at a superficial velocity of 12 mm/s, the Sauter mean diameter keeps on descending by degrees with the column height. This is mainly attributed to the fact that the breakup rate in the MSJBC was much higher than the coalescence rate because large bubbles are difficult to maintain in a stable existence in severe fluid turbulence. Bubble cutting is a gradual process, rather than the bubble moving through the mesh in a moment to complete the crushing process. In the case of the wire mesh, a noticeable change in the Sauter mean diameter was that its value underwent a significant reduction. The two stages mesh co-existed in the column and could further reduce the bubble size by cutting the bubble twice (see Figure 7), which is reflected in Figure 11 where the blue diamond crosses the blue line. Above the wire mesh, a larger number of bubbles, closer bubble spacing, and a higher interaction frequency was obtained, resulting in a higher coalescence rate and the bubble Sauter mean diameter in all cases experienced a gradual increase along the column height. A single mesh had the characteristic of a low cutting efficiency, and therefore the bubble diameter scatter plot is more loosely arranged than for the two wire meshes.

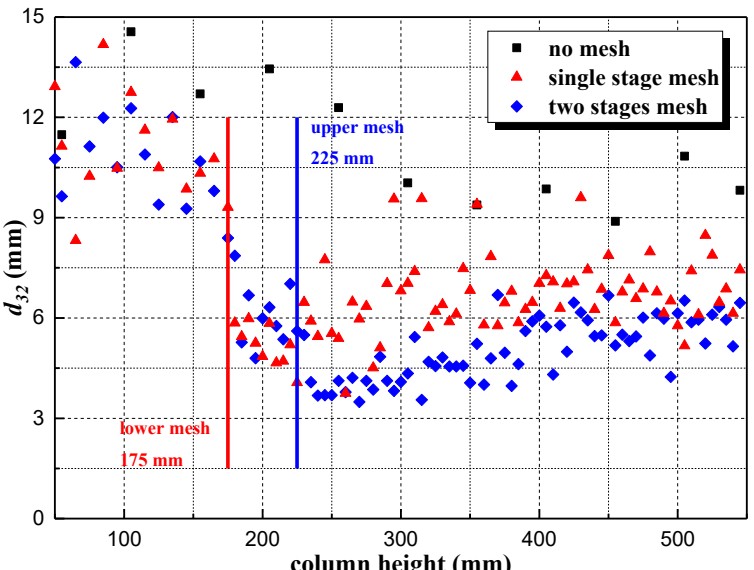

**Figure 11.** Variation of Sauter mean diameter $d_{32}$ along the height of the column equipped at a superficial velocity $U_g$ of 12 mm/s equipped with no (black square), single stage (red triangle) and two stages mesh (blue diamond) without chemisorption.

The overall Sauter mean diameter best represents the average size of bubbles in the column. The overall Sauter mean diameter in the column varied with the superficial gas velocity as shown in Figure 12. The results show that with an increasing superficial gas velocity, the bubble Sauter mean size increased (black line in Figure 12); however, as the conclusion described above, an increasing gas velocity improved the bubble diameter, but the dual effect of gas velocity caused by excessive gas velocity increased the instability of the large bubbles in turn. Therefore, when the apparent gas velocity increased from 4 to 12 mm/s, the overall bubble size showed the trend of a steady increase, but when the gas velocity further increased to 16 mm/s, the overall bubble size slightly decreased. The wire mesh cutting bubble played an important role in reducing the overall bubble size. In the cases investigated, the average bubble size decreased by 22.7% (single stage) and 29.7% (two stages), respectively, with the addition of wire mesh compared to without mesh. This further confirms that the two stages mesh arrangement can effectively reduce the size of the bubbles, and the multi-stage mesh arrangement can be considered in subsequent work to achieve the purpose of a mass transfer process intensification.

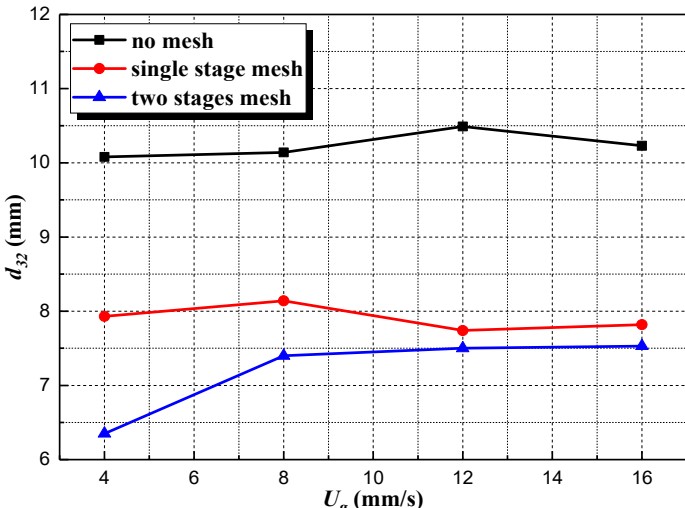

**Figure 12.** Overall Sauter mean diameter $d_{32}$ as a function of superficial gas velocity $U_g$ equipped with no, single stage and two stages mesh without chemisorption.

### 4.4. Effect of Different Mesh Configurations on Mass Transfer Performance

Gas holdup is defined as the fraction of gas in a certain amount of a gas–liquid mixture. The gas holdup can indirectly reflect the contact performance of the gas–liquid two-phase. From Figure 13, the superficial gas velocity ranged from 4 to 16 mm/s, as a function of gas velocity, and the gas holdup approximates a straight line. Through a linear fitting of the curve, the slope of the example without the wire mesh is 0.376. In the presence of the wire mesh, the bubble rising velocity reduced, the residence time prolonged, and the number of small bubbles increased caused by the intrusive effect, increasing the gas holdup. The slopes of adding a wire mesh and the two stages mesh were 0.382 and 0.389, respectively; however, there were more small bubbles in the condition of high gas velocity, so the growth of the gas holdup brought by the mesh only had a little influence. The average gas holdup increased by almost 5.7% for a single stage and 9.7% for two stages when the wire mesh was inserted.

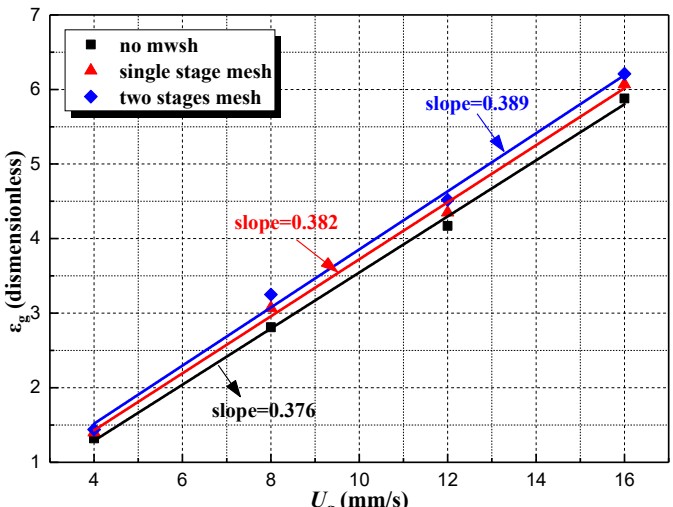

**Figure 13.** Gas holdup vs. superficial gas velocity $U_g$ (4~16 mm/s) for different mesh configurations without chemisorption.

Equation (4) denotes the relationship between the gas–liquid interfacial area and the gas holdup and Sauter mean diameter. The gas holdup and Sauter mean diameter are both functions of gas velocity, therefore, when processing the interfacial area, it can

be regarded as a function of the superficial gas velocity. The relationship between the gas–liquid interfacial area $a$ and the gas velocity growth is shown in Figure 14. Under the influence of the wire mesh, the gas holdup increased and the Sauter mean diameter decreased, so the gas–liquid interfacial area increased according to Equation (4). This means that the gas–liquid two-phase contact area was larger and the mass transfer area increased, which better enhances the mass transfer effect and achieves the purpose of enhancing the mass transfer. Compared with the case without the wire mesh, the interfacial area increased from 34.8% to 41.4% for the single mesh and from 43.5% to 73.2% for the two stages mesh.

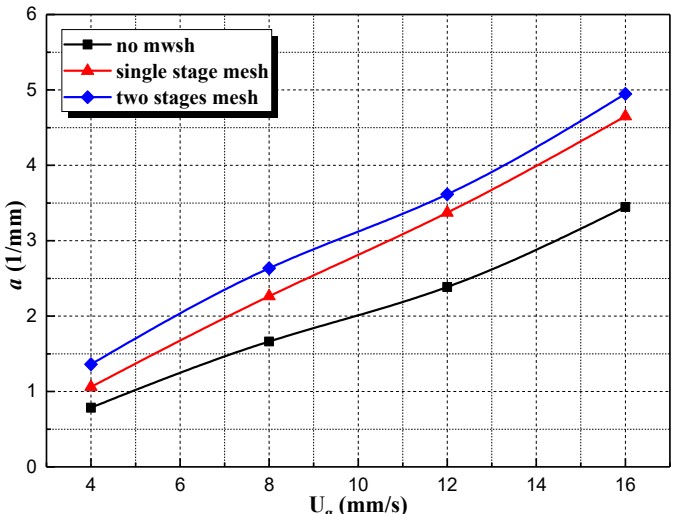

**Figure 14.** Interfacial area $a$ vs. superficial gas velocity $U_g$ (4~16 mm/s) for different mesh configurations without chemisorption.

A mass transfer experiment for carbon dioxide chemisorption into a sodium hydroxide solution in the MSJBC was designed to verify the enhanced effect of wire mesh on the mass transfer. Figure 15a represents the curve of the overall pH changes over time, and Figure 15b–d represents the curve of pH changes within a local time range. As shown in Figure 15a, the three curves had similar downward trends; however, the pH decay curve dropped more slowly in the absence of wire mesh than in the presence of the mesh. As the pH curve for the single stage and two stages mesh dropped fast to reach pH 7 in 300 s and 273 s, respectively, while the case with no mesh configuration took almost 351 s for reaching pH 7 at the same gas velocity of 12 mm/s. It can be seen from Figure 15b–d that the curve fitted once within the three representative linear pH drop ranges. The slope of the two stages mesh was greater than that of the single stage mesh and the non-wire mesh, while the slope of the single wire mesh was greater than without wire mesh. The results show that the wire mesh speeds up the pH drop of the solution within a limited time and shortens the reaction time. It can be concluded from the pH decay curve that a MSJBC with a mesh configuration performs much better than a MSJBC without an internal mesh configuration for chemical reactions limited by mass transfer efficiency.

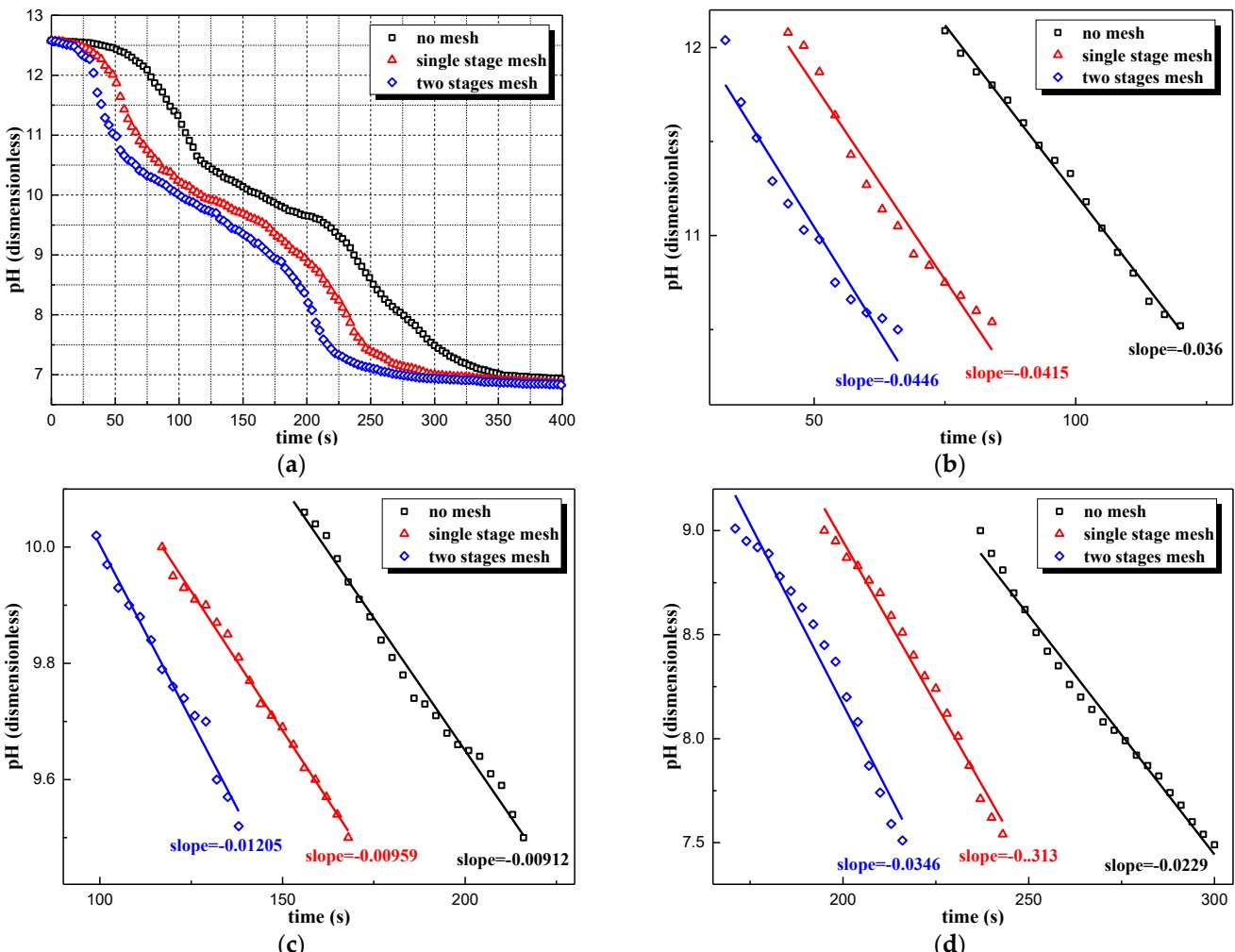

**Figure 15.** pH vs. time decay curve for a column at a superficial gas velocity $U_{g'}$ of 12 mm/s with various mesh configurations (no, single stage and two stages mesh) with chemisorption. (**a**) represents collating pH decay curve, (**b**) pH range: 12 to 10.5. (**c**) pH range: 10 to 9.5. (**d**) pH range: 9.0 to 7.5.

## 5. Conclusions

In this work, a micro-structured jet bubble column (MSJBC) was studied in detail equipped with different mesh configurations using digital image analysis for superficial gas velocities ranging from 4 to 16 mm/s. The effects of different mesh configurations (no, single stage and two wire mesh) on the bubble cutting, process intensification and chemical reactions were investigated experimentally. For the experiments, a non-invasive optical measurement method was applied to determine the bubble dynamics and to capture cutting images. The hydrodynamics of the MSJBC with different configurations such as those surrounding the flow field and liquid velocity were studied using an advanced particle image velocimetry (PIV) technique. To verify the effect of wire mesh on mass transfer enhancement, an experiment was carried out on the chemical absorption of $CO_2$ into an aqueous NaOH solution. The main experimental results are as follows:

(1) The bubble column performance with wire mesh was significantly better than that without wire mesh. In the presence of the mesh, the large bubbles were cut into small bubbles, and the average bubble size decreased by 22.7% (single stage) and 29.7% (two stages), respectively, which increased the gas–liquid interfacial area. The interaction between the wire mesh and the bubbles also enhanced the interface dynamics and updated the phase boundary, which meant a higher surface renewal rate and local mass transfer rate could be realized.

(2) The wire mesh affected the overall flow regime within the MSJBC and caused compartmentalized liquid circulation patterns above and below the mesh, resulting in less liquid back-mixing inside the MSJBC. The local vortex caused by the intrusive effect of the screen made it difficult for small bubbles to escape from the liquid phase and prolonged the gas residence time (reaction time).

(3) For the gas holdup and interfacial area, which are directly related to the mass transfer performance, the average gas holdup increased by almost 5.7% (single stage) and 9.7% (two stages), while the interfacial area increased from 34.8% to 41.4% for the single mesh and from 43.5% to 73.2% for the two stages mesh in the presence of the inserted mesh.

(4) Through chemisorption experiments of $CO_2$ into an NaOH aqueous solution, we concluded that the wire mesh could enhance chemical reactions subject to a poor mass transfer efficiency.

**Author Contributions:** Conceptualization, G.C. and Z.Z.; methodology, G.C. and Z.Z.; software, J.D.; validation, J.D. and G.C.; formal analysis, J.D. and F.G.; investigation, G.C. and Z.Z.; resources, Z.Z.; data curation, G.C.; writing—original draft preparation, Z.Z. and F.G.; writing—review and editing, G.C., J.D. and Z.Z; visualization, Z.Z. and J.D.; supervision, G.C., F.G. and J.D.; project administration, G.C., J.L. and J.D. All authors have read and agreed to the published version of the manuscript.

**Funding:** This work was supported by the Natural Science Foundation of Shandong Province (No.ZR2018BB071), Qingdao Science and Technology Plan Application Foundation Research Project (No.19-6-2-28-cg) and the Key Research and Development Project of Shandong Province (No.2019GSF109038).

**Institutional Review Board Statement:** Not applicable.

**Informed Consent Statement:** Not applicable.

**Data Availability Statement:** The data presented in this study are available on request from the corresponding author.

**Conflicts of Interest:** The authors declare no conflict of interest.

## Nomenclature

| | |
|---|---|
| $a$ | Gas–liquid interfacial area, $m^2/m^3$ |
| $C$ | bubble chord length, m |
| $d_{eq}$ | equivalent bubble diameter, m |
| $d_{max}$ | maximum bubble diameter, m |
| $d_{min}$ | minimum bubble diameter, m |
| $d_{32}$ | Sauter mean diameter, m |
| $D_C$ | bubble column diameter, m |
| $D_L$ | molecular diffusion coefficient, $m^2/s$ |
| $g$ | gravitational acceleration, $m/s^2$ |
| $h_f$ | final liquid height, m |
| $h_i$ | initial liquid height, m |
| $k_l\alpha$ | volumetric mass transfer coefficient, 1/s |
| $\bar{L}$ | displacement, m |
| $N$ | number of bubbles, dimensionless |
| $\Delta t$ | time, s |
| $\Delta \bar{t}$ | liquid velocity, m/s |
| $V_b$ | volume of the bubble |
| $U_g$ | superficial gas velocity, m/s |
| **Greek letters** | |
| $\varepsilon_g$ | gas holdup, dimensionless |

## Abbreviations

| | |
|---|---|
| BC | bubble column |
| LIF | laser induced fluorescence |
| MSJBC | micro-structured jet bubble column |
| NDF | number density function |
| PIV | particle image velocimetry |
| UDF | user-defined function |

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
