# Peer review of "Effect of Different Configurations on Bubble Cutting and Process Intensification in a Micro-Structured Jet Bubble Column Using Digital Image Analysis"

_processes, doi:10.3390/pr9122220_

Round 1
Reviewer 1 Report
The topic of the paper refers to an experimental study that was conducted in order to investigate the effect of different configurations on bubble cutting and process intensification in a micro-structured jet bubble column.
Generally, the manuscript is very interesting, the experimental procedures were well designed and the article is correctly written. Paper could be accepted for publication after a minor revision, taking into account the following suggestions:
Line 77 – „and a wire mesh” may be replaced with „one or two wire meshes” – because there are two different experiments
Line 145- It is not clear if the a is: „The gas-liquid interfacial area concentration ” or „The gas-liquid interfacial area” as presented in Introduction (Line 28) and in Nomenclature table
Figure 5d and figure 5e – are not discussed
Figure 6d and Figure 6e – may be addressed in Line 201: „will re-coalescence (see Figure 6c-6e)”
Lines 222 – 226 „ In this section, the effect of bubble cutting was examined and its mesh insertion on the surrounding flow field. The red dash line represents the position of the wire mesh, which is located at a distance of 175 mm (lower dash line) and 225 mm (upper dash line) away from the bottom gas nozzle. When investigating the influence of bubble cutting on the flow field, in order to observe the influence of single bubble cutting on the surrounding flow field and ignore the interference of surrounding bubbles as much as possible.” – must be rewritten to be understandable.
Line 253 – “And because the bubble trajectory is not vertical impact mesh” – is to be rewritten to be understandable.

Author Response
Manuscript ID: processes-1483549
Title: Effect of Different Configurations on Bubble Cutting, Process Intensification and Carbon Dioxide Mass Transfer in a Micro-structured Jet Bubble Column
Dear Ms.Jennie Zhu and reviewers,
Thank you very much for your email.
We would like to take this opportunity to thank you and the reviewers for the invaluable comments, which help us improve the manuscript greatly. We have revised the manuscript strictly according to the reviewer’s and editor’s comments. Together with the revised manuscript, here we also attach the Itemized Responses to all the reviewer’s and editor’s comments. We sincerely wish the manuscript is good enough to be accepted by this prestigious journal.
Sincerely Yours,
Jipeng Dong
Responses to all the Reviewers Comments
Reviewer #1:
The topic of the paper refers to an experimental study that was conducted in order to investigate the effect of different configurations on bubble cutting and process intensification in a micro-structured jet bubble column.
Generally, the manuscript is very interesting, the experimental procedures were well designed and the article is correctly written. Paper could be accepted for publication after a minor revision, taking into account the following suggestions.
【Author’s reply】: Thanks very much for the comments for our manuscript and for your recognition of our work. We believe that these comments can improve the quality of our manuscript. The replies to the specific comments are answered below.
Comment 1:
Line 77 – „and a wire mesh” may be replaced with „one or two wire meshes” – because there are two different experiments.
【Author’s reply 1】: Thanks for your suggestion, which points out our presentation error. In the section 2.1 in the revised manuscript, we accept your suggestion to modify the expression of the experimental device, and the results are discussed. The specific content is as follows:
“The experimental devices consist of a rectangular MSJBC (with dimensions of 100 mm width, 25 mm depth and 700 mm height) and one or two wire meshes (with 5.5 mm mesh opening and 0.7 mm wire diameter).”
Comment 2:
Line 145- It is not clear if the a is: „The gas-liquid interfacial area concentration ” or „The gas-liquid interfacial area” as presented in Introduction (Line 28) and in Nomenclature table.
【Author’s reply 2】: Thanks for your suggestion. In the manuscript, our description of "a" is inaccurate.“a” should represent the gas-liquid interfacial area rather than interfacial area concentration. We sincerely accept your suggestion that all the expressions about "a" in line 28, line 77 and in Nomenclature table in the manuscript should be changed to "gas-liquid interfacial area". The specific modifications are as follows:
“Bubble columns (BCs), which can be used as an efficient multiphase contactor and bioreactor [1,2], are usually used in many industrial applications for gas-liquid contacting processes, especially used in the case of high interfacial area a and intense gas-liquid mixing [3,4].”
“The gas-liquid interfacial area a is a function of gas holdup and d32, which can be calculated by the following formula:”
|
Nomenclature |
|
time, s |
|
|
a |
gas-liquid interfacial area, m2/m3 |
|
liquid velocity, m/s |
|
C |
bubble chord length, m |
Vb |
volume of the bubble |
|
deq |
equivalent bubble diameter, m |
Ug |
superficial gas velocity, m/s |
|
dmax |
maximum bubble diameter, m |
|
|
|
dmin |
minimum bubble diameter, m |
Greek letters |
|
|
d32 |
Sauter mean diameter, m |
εg |
gas holdup, dimensionless |
|
DC |
bubble column diameter, m |
|
|
|
DL |
molecular diffusion coefficient, m2/s |
Abbreviations |
|
|
g |
gravitational acceleration, m/s2 |
BC |
bubble column |
|
hf |
final liquid height, m |
LIF |
laser induced fluorescence |
|
hi |
initial liquid height, m |
MSJBC |
micro-structured jet bubble column |
|
klα |
volumetric mass transfer coefficient, 1/s |
NDF |
number density function |
|
|
displacement, m |
PIV |
particle image velocimetry |
|
N |
number of bubbles, dimensionless |
UDF |
user-defined function |
Comment 3:
Figure 5d and figure 5e – are not discussed.
【Author’s reply 3】: Thanks for your suggestion. In describing Figure 5 in section 4.1 in the manuscript, we omitted a detailed description of each of these figures. In the revised manuscript, we re-described the image contained in Figure 5. The specific content is as follows:
“From Figure 5a-5c, bubble cutting occurs in the presence of the mesh. As shown in Figure 5d and 5e, leading bubbles are effectively cut into numerous daughter bubbles, and the daughter bubbles are dispersed due to the invasion effect of the wire mesh, avoiding the recoalescence during the ascent process.”
Comment 4:
Figure 6d and Figure 6e – may be addressed in Line 201: „will re-coalescence (see Figure 6c-6e)”.
【Author’s reply 4】: Thanks for your suggestion. We heartily accept your suggestion. According to your suggestion, we have revised "Figure 6c" into "Figure 6c-6e" in section 4.1 in the revised manuscript. The specific content is as follows:
“Large aggregation bubbles in the cutting process are not completely cut and broken, and part of bubbles passing through the wire mesh will re-coalescence (see Figure 6c-6e).”
Comment 5:
Lines 222 – 226 „ In this section, the effect of bubble cutting was examined and its mesh insertion on the surrounding flow field. The red dash line represents the position of the wire mesh, which is located at a distance of 175 mm (lower dash line) and 225 mm (upper dash line) away from the bottom gas nozzle. When investigating the influence of bubble cutting on the flow field, in order to observe the influence of single bubble cutting on the surrounding flow field and ignore the interference of surrounding bubbles as much as possible.” – must be rewritten to be understandable.
【Author’s reply 5】: Thanks for your suggestion. In section 4.2 in the revised manuscript, in response to your question, we have carefully revised the above problematic expressions according to your comments for easy understanding. The revised content is as follows:
“In this section, the intrusive effect of the inserted wire mesh on the surrounding flow field was examined. In Figure 8, the red horizontal dashed line indicates the location of the wire mesh. The instantaneous flow field distribution obtained after post-processing is shown in Figure 8. Figures 8a and 8c are original images, and 8b and 8d are correspondingly processed flow field information images. From Figure 8d and 8e, the distribution of the flow field is scattered and chaotic, indicating that the gas at low superficial gas velocity does not play a role in regulating the flow field. The flow field distribution around the cut bubbles is divergent, which can be devoted to dispersing bubbles, increasing the distance between bubbles and reducing the coalescence rate of bubbles. There are small eddies resting in the path of the bubbles. The dissipation of a small vortex results in a disorderly change in the flow field. The flow field around the bubbles after cutting will produce local vortices under the influence of discrete bubbles. In the presence of two stages mesh, the intrusive effect of wire mesh increases, the flow field suffers stronger interference, and the chaotic degree of the flow field increases. Some small local vortex is formed between the mesh.”
Comment 6:
Line 253 – “And because the bubble trajectory is not vertical impact mesh” – is to be rewritten to be understandable for easier comprehension.
【Author’s reply 6】: Thanks for your suggestion. We have reedited the ambiguous sentence. The revised content is as follows:
“And because of the influence of the curved bubble trajectory, the bubble is not perpendicularly through the mesh but with the mesh is a certain Angle. The bubble cutting caused by this non-vertical form through the mesh will increase the radial movement of the bubble, which is not conducive to maintaining the stability of the flow field.”
Comment 7:
English language and style: Moderate English changes required.
【Author’s reply 7】: Thanks for your suggestion. We apologize for any problems with English grammar and structure in the submitted manuscript. In the revised manuscript, we have carefully revised the grammatical errors and typos of the whole manuscript. In addition, we readjusted the structure of the text. We have asked several colleagues who have backgrounds abroad checked and refined the language of the manuscript. We wish that the language is now acceptable for the publication.

Reviewer 2 Report
Most of the data seems good and the conclusion expressed in the text seems supported. This article would be useful for people working with CFD and population balance since the geometry of the bubble column is simple. I don't see many problems with the method and the results obtained.
This being said, a lot of work has been done on the bubble column and I know that many teams use them to support their research in CFD. Most of the techniques and phenomena involved are known and the whole novelty of the article relies on the data itself. That gives me the impression that the document lacks a bit of originality.
The English and the structure of the text hinder the reading and make it much harder than necessary to follow. I don't think any work has to be added, but the article necessitates a good polish, which will ultimately make the communication much better and the text more impactful.
I would reconsider after a good revision.
Author Response
Manuscript ID: processes-1483549
Title: Effect of Different Configurations on Bubble Cutting, Process Intensification and Carbon Dioxide Mass Transfer in a Micro-structured Jet Bubble Column
Dear Ms.Jennie Zhu and reviewers,
Thank you very much for your email.
We would like to take this opportunity to thank you and the reviewers for the invaluable comments, which help us improve the manuscript greatly. We have revised the manuscript strictly according to the reviewer’s and editor’s comments. Together with the revised manuscript, here we also attach the Itemized Responses to all the reviewer’s and editor’s comments. We sincerely wish the manuscript is good enough to be accepted by this prestigious journal.
Sincerely Yours,
Jipeng Dong
Responses to all the Reviewers Comments
Reviewer #2:
Comment 1:
Most of the data seems good and the conclusion expressed in the text seems supported. This article would be useful for people working with CFD and population balance since the geometry of the bubble column is simple. I don't see many problems with the method and the results obtained.
【Author’s reply1】: Thank you very much for your comments and affirmation of our work. As stated in your comments, the experimental results of our work can provide data support for the development of CFD-PBM coupled model. In fact, after the in-depth discussion of several of our co-authors, we plan to take this constructive suggestion as the next key research content.
Comment 2:
This being said, a lot of work has been done on the bubble column and I know that many teams use them to support their research in CFD. Most of the techniques and phenomena involved are known and the whole novelty of the article relies on the data itself. That gives me the impression that the document lacks a bit of originality.
【Author’s reply2】: Thank you very much for pointing out the shortcomings of our article. The main purpose of this work is to extend our previous research [1-3], thus we have made a more detailed description of the known phenomenon, and made more measurements of its experimental phenomenon, and provided more experimental data. Our experimental work makes up for the qualitative research on experimental phenomena in previous studies, and we provide quantitative evidence of experimental phenomena through a large number of data. In future work, we will combine CFD with population balance model (PBM) to increase the innovation and originality of our work.
[1] Wang, W.W.; Li, S.Y.; Li, J.L. Experimental determination of bubble size distributions in laboratory scale sieve tray with mesh. Ind. Eng. Chem. Res. 2012, 51, 7067–7072.
[2] Chen, G.H.; Zhu, H.T.; Guo, X.L.; Wang, W.W.; Li, J.L. Flow characteristics and CFD simulation on a high-efficiency sieve tray with bubble crusher. CIESC. Journal. 2017, 68, 4633–4640.
[3] Li, X.; Wang, W.W.; Zhang, P.; Li, J.L.; Chen, G.H. Interactions between gas–liquid mass transfer and bubble behaviours. Roy. Soc. Open. Sci. 2019, 6, 190136.
Comment 3:
The English and the structure of the text hinder the reading and make it much harder than necessary to follow. I don't think any work has to be added, but the article necessitates a good polish, which will ultimately make the communication much better and the text more impactful.
【Author’s reply3】: Thanks very much for your suggestion. We apologize for any problems with English grammar and structure in the submitted manuscript. In the revised manuscript, we have carefully revised the grammatical errors and typos of the whole manuscript. In addition, we readjusted the structure of the text. We have asked several colleagues who have backgrounds abroad checked and refined the language of the manuscript. We wish that the language is now acceptable for the publication.
Round 2
Reviewer 2 Report
The English is improved and the modification made to the introduction put the work in value a bit more. It a pass for me. Congrats!